# EIDETIC 3D LSTM:
# A MODEL FOR VIDEO PREDICTION AND BEYOND

**Yunbo Wang**[1*]**, Lu Jiang**[2]**, Ming-Hsuan Yang**[2,3]**, Li-Jia Li**[4]**, Mingsheng Long**[1]**, Li Fei-Fei**[4]
[1]Tsinghua University, [2]Google AI, [3]University of California, Merced, [4]Stanford University

## ABSTRACT

Spatiotemporal predictive learning, though long considered to be a promising self-supervised feature learning method, seldom shows its effectiveness beyond future video prediction. The reason is that it is difficult to learn good representations for both short-term frame dependency and long-term high-level relations. We present a new model, *Eidetic 3D LSTM* (E3D-LSTM), that integrates 3D convolutions into RNNs. The encapsulated 3D-Conv makes local perceptrons of RNNs motion-aware and enables the memory cell to store better short-term features. For long-term relations, we make the present memory state interact with its historical records via a gate-controlled self-attention module. We describe this memory transition mechanism *eidetic* as it is able to effectively recall the stored memories across multiple time stamps even after long periods of disturbance. We first evaluate the E3D-LSTM network on widely-used future video prediction datasets and achieve the state-of-the-art performance. Then we show that the E3D-LSTM network also performs well on the early activity recognition to infer what is happening or what will happen after observing only limited frames of video. This task aligns well with video prediction in modeling action intentions and tendency.

## 1 INTRODUCTION

A fundamental problem in spatiotemporal predictive learning is how to effectively learn good representations for video inference or reasoning. Currently, recurrent neural networks (RNNs) remain to be the most promising models in this field, and have achieved state-of-the-art results on a number of future video prediction benchmarks (Wang et al., 2018b; Oliu et al., 2018). However, beyond frames prediction, RNN based models are less effective in learning high-level video representations or capturing long-term relations. On the other hand, recent studies demonstrate that 3D Convolutional Neural networks (3D-CNNs) surpass RNNs in learning better representations for action classification (Carreira & Zisserman, 2017; Tran et al., 2015). For instance, variants of 3D-CNNs, such as Inflated 3D-CNNs, have significantly increased action classification accuracy over the UCF 101 and Kinetics datasets. These 3D-CNN architectures have no recurrent structures but instead employ 3D convolution (3D-Conv) and 3D pooling operations to preserve temporal information of the input sequences which would be otherwise discarded in classical 2D convolution operations.

Motivated by the recent success of 3D-CNNs, in this paper we propose a new model for spatiotemporal predictive learning based on both recurrent modeling (for temporal dependency) and feed-forward 3D-Conv modeling (for local dynamics). A plausible approach, of course, is to simply stack 3D-Convs and each RNN unit in a feed-forward way using 3D-Convs for either perceiving fine-grained features from raw videos or combining high-level representations. However, as shown in our experiments, these straightforward extensions may not outperform the baseline RNN model. We attribute these findings to that RNNs and 3D-CNNs represent two very different mechanisms for the same purpose of spatiotemporal modeling, and connecting them directly fails to exploit their complementary advantages. Therefore, it remains challenging and requires principled approaches to design an effective spatiotemporal network.

To this end, we propose a new model called *Eidetic 3D LSTM* (E3D-LSTM) for spatiotemporal predictive learning. We introduce an *eidetic 3D memory* to: a) memorize local appearance and motion in a short spatiotemporal volume, and b) recall the long-range historical context by learning

---

[*]Corresponding author: `wangyb15@mails.tsinghua.edu.cn`. Work done in part at Google AI.

to attend to previous memory states. Regarding the *short-term dependency*, in many cases, spatiotemporal predictive modeling mainly depends on temporally nearby appearances and on-going short-term motions. All the information is encapsulated into the eidetic 3D memory cell with a short time convolution window, and used in recurrent transitions. Our experimental results show that integrating 3D-Conv deep into RNNs is effective for modeling local representations in a consecutive manner. On the other hand, for *long-term interactions*, which is important for predicting non-stationary or periodical videos as well as learning high-level video representations, we exploit a self-attention mechanism controlled by revised recurrent gates to recall temporally distant memory. The current memory state of E3D-LSTM is learned to attend to all previous relevant moments. Our experimental results verify that this attention mechanism is beneficial for long-term memorization. We describe this memory transition mechanism *eidetic* as it is able to effectively recall the stored memories across multiple time stamps even after long periods of disturbance.

To the best of our knowledge, the proposed E3D-LSTM model is among the first approaches that leverage 3D-Conv in RNNs. We empirically validate it on standard spatiotemporal predictive tasks and an early activity recognition task over four benchmarks: a) on future video prediction, it achieves the best-published accuracy on three classical benchmarks; b) on early activity recognition, it outperforms the state-of-the-art action recognition methods. In addition, we show that self-supervised learning can further improve the performance of early activity recognition. We present ablation studies to verify the effectiveness of all modules in the proposed E3D-LSTM model.

## 2 RELATED WORK AND PROBLEM CONTEXT

**Spatiotemporal Predictive Learning Models.** In recent years, RNNs have been extensively used in sequence prediction and future frame prediction. Srivastava et al. (2015) extended the LSTM-based sequence to sequence model (Sutskever et al., 2014) for language modeling to learning video representations. Shi et al. (2015) proposed the convolutional LSTM by integrating convolutions into recurrent state transitions for high-dimensional sequence prediction. The convolutional LSTM model is extended by Finn et al. (2016) to predict future states of robotic environments. Villegas et al. (2017) leveraged optical flow to help capture short-term video dynamics for video prediction. Xu et al. (2018) proposed a two-stream RNN that deals with structural video content in separate streams. Kalchbrenner et al. (2017) introduced a sophisticated model that extends recurrent structures to estimate local dependencies between adjacent pixels. While this video pixel network (VPN) model is able to describe image sequences, the computational load is prohibitively high.

The above-mentioned recurrent models predict future frame mainly based on sequentially updated memory states. When the memory cell is refreshed, older memories will be discarded immediately. In contrast, the proposed E3D-LSTM model maintains a list of historical memory records and revokes them when necessary, thereby facilitating long-range video reasoning. While in spirit this idea is similar to the self-attention module in feed-forward networks (Vaswani et al., 2017; Wang et al., 2018a), we exploit it to correlate long-term and short-term video representations in this work.

Another significant difference between the above-mentioned prior work and the proposed model is that we use 3D-Convs as basic operations inside the E3D-LSTM instead of fully-connected or 2D convolution operations. We show using 3D-Convs to model recurrent state-to-state transitions can significantly improve prediction performance. This idea is motivated by recent advances in video classification (a high-level representation learning task) (Ji et al., 2013; Tran et al., 2015; Carreira & Zisserman, 2017). We note that Vondrick et al. (2016) and Tulyakov et al. (2018) also introduced 3D-CNNs for spatiotemporal predictive learning. However, these networks are feed-forward and do not capture temporal consistency effectively.

Future prediction errors of an imperfect model can be categorized by two factors: a) the "systematic errors" caused by a lack of modeling ability to the deterministic variations; b) the stochastic, inherent uncertainty of the future. We aim to minimize the first factor in this work. For the second factor, numerous methods have applied adversarial training or variational auto-encoders to video prediction, for example (Mathieu et al., 2016; Vondrick et al., 2016; Denton & Fergus, 2018; Bhattacharjee & Das, 2017; Tulyakov et al., 2018; Lu et al., 2017; Wichers et al., 2018).

**Convolutional Recurrent Networks.** Our model is closely related to convolutional recurrent networks. In the ConvLSTM network (Shi et al., 2015), all state transitions are implemented with 2D convolutions. As such, the transition function is no longer *permutation invariant* and able to better

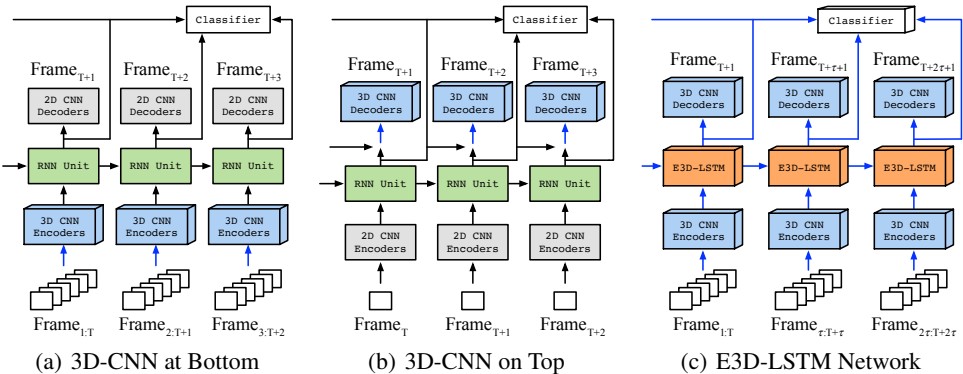

(a) 3D-CNN at Bottom    (b) 3D-CNN on Top    (c) E3D-LSTM Network

Figure 1: Three approaches to integrate 3D-Convs into recurrent networks. Blue arrows indicate data transition paths with 3D-Convs (for feed-forward features or recurrent hidden states). The diagrams are simplified for illustration, with fewer layers and RNN states than what are actually used in our experiments. The classifiers are removed when being trained for future video prediction.

perceive relations in a spatiotemporal neighborhood. The spatiotemporal LSTM (ST-LSTM) is characterized by delivering two memory states separately (Wang et al., 2017): memory $\mathcal{M}$ in a zigzag direction and memory $\mathcal{C}$ being passed horizontally (see Appendix A for details). In this model, $\mathcal{M}$ provides greater capability to model short-term motions, and $\mathcal{C}$ is adopted from fully-connected LSTMs (Hochreiter & Schmidhuber, 1997) to ease the vanishing gradient problem. Although the ST-LSTM performs well on video prediction benchmarks, it does not capture long-term video relations effectively. The forget gates of memory $\mathcal{C}$ tend to respond strongly to short-term features, thereby easily falling into a saturated zone (with values between $0$ and $0.1$) and interrupting long-range information flows. We adopt the zigzag updating route of memory $\mathcal{M}$ from the ST-LSTM, while improving the forgetting mechanism in updating the temporal memory $\mathcal{C}$. We also increase the dimensions of memory states and take 3D-Convs as the basic operators for state transitions.

## 3 EIDETIC 3D LSTM

This section first presents the Eidetic 3D LSTM for perceiving and memorizing both short-term and long-term representations in videos. We then discuss a scheduled multi-task learning strategy that uses predictive learning as an auxiliary self-supervised task for activity recognition.

### 3.1 3D CONVOLUTIONS IN RECURRENT NETWORK

An ideal predictive model relies on effective learning of video representations. RNNs and 3D-CNNs are network architectures of different mechanisms for modeling spatiotemporal data. In this work, we aim to leverage the strength of each one in a unified architecture and start the discussion with two plausible extensions of stacking 3D-Convs and RNN units. Figure 1(a) and 1(b) illustrate two hybrid baseline networks which add 3D-CNNs before or after stacked spatiotemporal LSTMs.

However, we find that integrating the 3D-Convs outside the LSTM unit performs noticeably worse than the baseline RNN model. To this end, we propose a "deeper" integration of 3D-Convs inside the LSTM unit in order to incorporate the convolutional features into the recurrent state transition over time. Figure 1(c) shows the overall encoder-decoder architecture. In this model, a consecutive of $T$ input frames are first encoded by a few layers of 3D-Convs to obtain high-dimension feature maps. The 3D-Conv feature maps are directly fed into a novel E3D-LSTM to model the long-term spatiotemporal interaction. Finally, the E3D-LSTM hidden states are decoded by a number of stacked 3D-Conv layers to get the predicted video frames. For classification tasks, the hidden states can be directly used as the learned video representation.

### 3.2 EIDETIC MEMORY TRANSITION

The architecture of the proposed Eidetic 3D LSTM is illustrated in Figure 2, where the red arrows indicate short-term information flow and the blue arrows denote long-term information flow. There are 4 inputs: $\mathcal{X}_t$, the 3D-Conv feature maps from encoders or hidden states from the previous E3D-

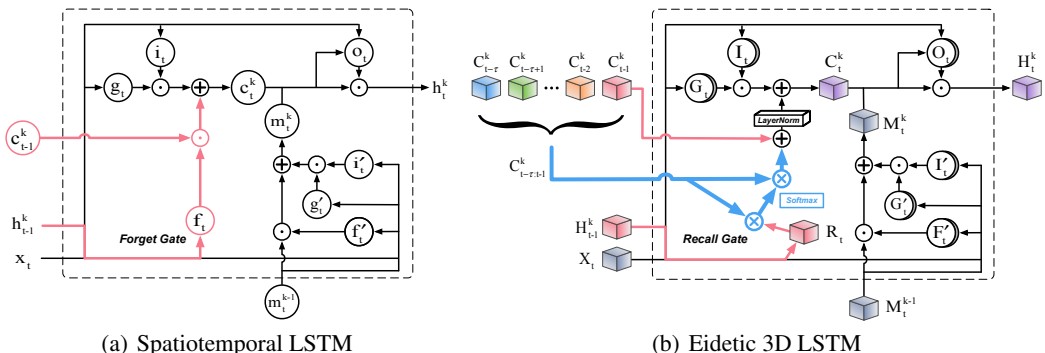

(a) Spatiotemporal LSTM           (b) Eidetic 3D LSTM

Figure 2: Comparison of (a) the standard memory transition approach in the Spatiotemporal LSTM and (b) the attentive memory transition approach in the Eidetic 3D LSTM. Red arrows indicate the short-term information flow. Blue arrows are the attentive memory flow, which potentially enables our model to capture the long-term relations. Cubes denote higher-dimensional hidden states and memory states. Cylinders denote higher-dimensional gates. $\odot$ is the Hadamard product. $\otimes$ is the matrix product after reshaping matrices into appropriate 2-dimensional forms.

LSTM layer; $\mathcal{H}_{t-1}^k$, the hidden states from previous time stamp; $\mathcal{C}_{t-1}^k$, the memory states from previous time stamp; and $\mathcal{M}_t^{k-1}$, the previous spatiotemporal memory states described earlier.

We use recurrent 3D-Convs as motion-aware perceptrons to extract short-term appearance and local motions in continuous space-time fields and store them in a small spatiotemporal volume. As such, video appearance and short-term motions can be encoded in $\mathbb{R}^{T \times H \times W \times C}$ tensors, in which each dimension indicates temporal depth, spatial size, and the number of feature map channels, respectively. By inflating the memory state along the time dimension, we found that the proposed E3D-LSTM becomes more capable of characterizing and memorizing local or short-term motions.

To capture the long-term frame interactions, we improve the recurrent transition function of the memory states by proposing a new memory RECALL mechanism:

$$
\begin{aligned}
\mathcal{R}_t &= \sigma(W_{xr} * \mathcal{X}_t + W_{hr} * \mathcal{H}_{t-1}^k + b_r) \\
\mathcal{I}_t &= \sigma(W_{xi} * \mathcal{X}_t + W_{hi} * \mathcal{H}_{t-1}^k + b_i) \\
\mathcal{G}_t &= \tanh(W_{xg} * \mathcal{X}_t + W_{hg} * \mathcal{H}_{t-1}^k + b_g) \\
\text{RECALL}(\mathcal{R}_t, \mathcal{C}_{t-\tau:t-1}^k) &= \text{softmax}(\mathcal{R}_t \cdot (\mathcal{C}_{t-\tau:t-1}^k)^{\mathsf{T}}) \cdot \mathcal{C}_{t-\tau:t-1}^k \\
\mathcal{C}_t^k &= \mathcal{I}_t \odot \mathcal{G}_t + \text{LayerNorm}(\mathcal{C}_{t-1}^k + \text{RECALL}(\mathcal{R}_t, \mathcal{C}_{t-\tau:t-1}^k)),
\end{aligned}
\tag{1}
$$

where $\sigma$ is the sigmoid function, $*$ is the 3D-Conv operation, $\odot$ is the Hadamard product, $\cdot$ is the matrix product after reshaping the recall gate $\mathcal{R}_t$ and memory states $\mathcal{C}_{t-\tau:t-1}^k$ into $\mathbb{R}^{THW \times C}$ and $\mathbb{R}^{\tau THW \times C}$ matrices, respectively, and $\tau$ is the number of memory states that are concatenated along the temporal dimension. Three terms are involved in computing $\mathcal{C}_t^k$. The first one $\mathcal{I}_t \odot \mathcal{G}_t$ encodes local video appearance and motions, where $\mathcal{I}_t$ is the input gate and $\mathcal{G}_t$ is the input modulation gate like standard LSTMs. The second one $\mathcal{C}_{t-1}^k$ can be viewed as a short-cut connection from the previous memory state, which captures short-term changes between adjacent time stamps. In this process, the accessible memory field is fixed and limited. Therefore, we introduce the third term of memory transition function, modeling long-term video relations according to local motion and appearance (as encoded in $\mathcal{X}_t$ and $\mathcal{H}_{t-1}^k$). The RECALL function is implemented as an attentive module to compute the relationship between the encoded local patterns and the whole memory space. A set of parameterized gates $\mathcal{R}_t$, acting as memory access instructions, control where and what to attend in historical memory records. These two terms are respectively designed for short-term and long-term video modeling. We integrate them in a unified network by applying layer normalization (Ba et al., 2016) to their element-wise sum, in order to mitigate the covariant shift and stabilize the training process, as it has been commonly used in RNNs. The hyper-parameter $\tau$ in $\mathcal{C}_{t-\tau:t-1}^k$ decides how many historical memory states are attended by the recall gate $\mathcal{R}_t$. To involve more long-term relations, in most experiments, we take $\mathcal{C}_{1:t-1}^k$ as the inputs of the RECALL function and do not fix $\tau$. Whereas in particular, we enable online recognition by setting $\tau$ to 5.

Unlike the conventional memory transition function, the RECALL function learns the size of temporal interactions. For longer sequences, this allows attending to distant states containing salient in-

formation. Our work is partially motivated by self-attention mechanisms (Lin et al., 2017; Vaswani et al., 2017). However, in our model, the attention mechanism is not applied over the output states but during the memory transitions. It is used to evoke past memories from distant time stamps for memorizing and distilling useful information from what has been perceived. We show that learning attention over previous memory states is beneficial in recalling the long-range historical context. The memory tensor $\mathcal{C}_t^k$ is named *eidetic 3D memory* and the entire unit is called E3D-LSTM. We also exploit the same RECALL method to correlate $\mathcal{M}_t^{1:k}$ along the vertical memory transition flow, but it turns out to be less helpful. With the updated memory state $\mathcal{C}_t^k$, the output hidden states are:

$$
\begin{aligned}
\mathcal{I}_t' &= \sigma(W_{xi}' * \mathcal{X}_t + W_{mi} * \mathcal{M}_t^{k-1} + b_i') \\
\mathcal{G}_t' &= \tanh(W_{xg}' * \mathcal{X}_t + W_{mg} * \mathcal{M}_t^{k-1} + b_g') \\
\mathcal{F}_t' &= \sigma(W_{xf}' * \mathcal{X}_t + W_{mf} * \mathcal{M}_t^{k-1} + b_f') \\
\mathcal{M}_t^k &= \mathcal{I}_t' \odot \mathcal{G}_t' + \mathcal{F}_t' \odot \mathcal{M}_t^{k-1} \\
\mathcal{O}_t &= \sigma(W_{xo} * \mathcal{X}_t + W_{ho} * \mathcal{H}_{t-1}^k + W_{co} * \mathcal{C}_t^k + W_{mo} * \mathcal{M}_t^k + b_o) \\
\mathcal{H}_t^k &= \mathcal{O}_t \odot \tanh(W_{1\times1\times1} * [\mathcal{C}_t^k, \mathcal{M}_t^k]),
\end{aligned}
\tag{2}
$$

where $W_{1\times1\times1}$ is the $1 \times 1 \times 1$ convolutions for the transformation of the channel number. $\mathcal{I}_t'$, $\mathcal{G}_t'$, and $\mathcal{F}_t'$ are gate structures of the spatiotemporal memory. $\mathcal{O}_t$ is the output gate.

### 3.3 Self-supervised Auxiliary Learning

For many supervised tasks such as video action recognition, there are often not enough supervisions or annotations over time for training a satisfactory RNN. As an auxiliary measure to this problem, future video prediction is considered as a promising representation learning approach that is more densely supervised over time and might extract useful features to assist video understanding.

We consider two tasks: the pixel-level future frames prediction and another video-level classification task (early activity recognition in our case). For frames prediction, the objective function is:

$$
\mathcal{L}_{\text{prediction}} = \|\mathcal{X} - \widehat{\mathcal{X}}\|_F^2 + \|\mathcal{X} - \widehat{\mathcal{X}}\|_1,
\tag{3}
$$

where $\widehat{\mathcal{X}}$ and $\mathcal{X}$ are respectively predicted and ground truth future frames. $\|\cdot\|_F$ is the Frobenius norm. For early activity recognition, we make the models for these two tasks share the same network backbone in the end-to-end training using a multi-task learning objective:

$$
\mathcal{L}_{\text{recognition}} = \lambda\|\mathcal{X} - \widehat{\mathcal{X}}\|_F^2 + \mathcal{L}_{\text{ce}}(\mathcal{Y}, \widehat{\mathcal{Y}}),
\tag{4}
$$

where $\widehat{\mathcal{Y}}$ and $\mathcal{Y}$ are high-level predictions and corresponding ground truth classes. $\mathcal{L}_{\text{ce}}$ is the cross-entropy loss for classification, and $\lambda$ is the weight factor.

Although improving both tasks requires proper long short-term contextual representations, there is no guarantee that features learned with pixel-level supervisions will fully align with any high-level objectives. We thus introduce a scheduled learning strategy where the objective function is gradually inclined from one task to the other in a curriculum learning manner (Bengio et al., 2009). Specifically, we apply a linear decay to $\lambda$ over the number of iterations $i$:

$$
\lambda(i) = \max(\eta, \lambda(0) - \epsilon \cdot i),
\tag{5}
$$

where $\lambda(0)$ and $\eta$ are respectively maximum and minimum values of $\lambda(i)$, $\epsilon$ controls the decreasing speed of the role of the auxiliary task. We call this approach the *Self-supervised Auxiliary Learning*.

## 4 Experiments

We evaluate the proposed E3D-LSTM model on two tasks: future video prediction and early activity recognition. These two tasks are of great importance with numerous applications that require effective spatiotemporal predictive models. We demonstrate that the E3D-LSTM model performs favorably against the state-of-the-art models on four challenging datasets. The source code and trained models will be made available to the public.

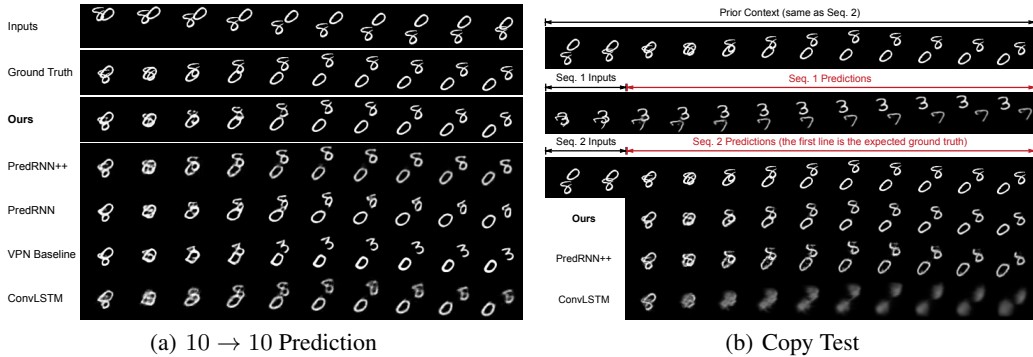

(a) $10 \rightarrow 10$ Prediction                    (b) Copy Test

Figure 3: Video prediction examples on the Moving MNIST dataset.

Table 1: Results on the Moving MNIST dataset. All models, except DFN and VPN, are trained with a comparable number of parameters. Higher SSIM or lower MSE scores indicate better results.

| MODEL | $10 \rightarrow 10$ | | COPY | |
|---|---|---|---|---|
| | SSIM | MSE | SSIM | MSE |
| CONVLSTM (SHI ET AL., 2015) | 0.713 | 96.5 | 0.539 | 143.2 |
| DFN (DE BRABANDERE ET AL., 2016) | 0.726 | 89.0 | 0.598 | 153.9 |
| CDNA (FINN ET AL., 2016) | 0.728 | 84.2 | 0.671 | 127.1 |
| FRNN (OLIU ET AL., 2018) | 0.819 | 68.4 | 0.694 | 110.5 |
| VPN BASELINE (KALCHBRENNER ET AL., 2017) | 0.870 | 64.1 | 0.736 | 78.0 |
| PREDRNN (WANG ET AL., 2017) | 0.869 | 56.5 | 0.745 | 80.3 |
| PREDRNN++ (WANG ET AL., 2018B) | 0.885 | 46.3 | 0.807 | 69.9 |
| E3D-LSTM | **0.910** | **41.3** | **0.852** | **56.8** |

## 4.1 FUTURE VIDEO PREDICTION: MOVING MNIST

We first evaluate the E3D-LSTM model against the state-of-the-art video prediction models on a commonly used synthetic benchmark dataset with moving digits. All experiments are conducted using TensorFlow (Abadi et al., 2016) and trained with the ADAM optimizer (Kingma & Ba, 2015) to minimize the $l_1 + l_2$ loss over every pixel in the frame. For fair comparisons, we ensure all models to have comparable numbers of parameters, and apply the same scheduled sampling strategy (Bengio et al., 2015) in order to reduce the difficulty of training recurrent models.

**Dataset and Setup.** The moving MNIST dataset is constructed by randomly sampling two digits from the original MNIST dataset and making them float and bounce at boundaries with a constant velocity and angle inside a black canvas of $64 \times 64$ pixels. The whole dataset has a fixed number of entries, $10,000$ sequences for training, $3,000$ for validation and $5,000$ for test.

We stack $4$ E3D-LSTMs in the architecture illustrated in Figure 1(c), leaving out 3D-CNN encoders for this task. To retain the shape of hidden states over time, the integrated 3D-Conv operators are composed of a $2 \times 5 \times 5$ (time $\times$ height $\times$ width) convolutions and a corresponding transposed convolution with the same filter size. The number of hidden state channels of each E3D-LSTM is $64$. The temporal stride is set to $1$ and there is one overlapping frame over consecutive time stamps. A single 3D-Conv layer is used as the decoder to map motion-aware hidden states to output frames.

The E3D-LSTM model is evaluated against the state-of-the-art methods including the ConvLSTM network (Shi et al., 2015), DFN (De Brabandere et al., 2016), CDNA (Finn et al., 2016), VPN baseline model with CNN decoders (Kalchbrenner et al., 2017), PredRNN (Wang et al., 2017), PredRNN++ (Wang et al., 2018b) and FRNN (Oliu et al., 2018).

**Main Results.** Table 1 shows the performance of the evaluated models using a common setting in the literature: generating 10 future frames given the previous 10 observations (denoted as $10 \rightarrow 10$). We use the per-frame structural similarity index measure (SSIM) (Wang et al., 2004) and per-frame mean squared error (MSE) for evaluation. The SSIM ranges between $-1$ and $1$, representing the similarity between the generated image and the ground truth. As shown in the second column ($10 \rightarrow 10$) of Table 1, our model performs well against the state-of-the-art methods in both metrics. The results show that the E3D-LSTM network is effective in modeling spatiotemporal data for video

Table 2: Ablation study on the Moving MNIST dataset ($10 \rightarrow 10$).

| MODEL | SSIM | MSE |
|---|---|---|
| BASELINE 1: 3D-CNN AT BOTTOM (FIGURE 1(A)) | 0.859 | 50.6 |
| BASELINE 2: 3D-CNN ON TOP (FIGURE 1(B)) | 0.862 | 53.4 |
| BASELINE 3: OURS (W/O 3D CONVOLUTIONS) | 0.894 | 44.2 |
| BASELINE 4: OURS (W/O MEMORY ATTENTION) | 0.880 | 45.7 |
| E3D-LSTM | **0.910** | **41.3** |

prediction. Figure 3(a) shows the qualitative comparisons in which our model predicts future frames from entangled digits better than other methods.

**Copy Test.**    We evaluate the proposed model using the *Copy Test* setting where the task is to memorize useful information in a longer input sequence when the recurrent disturbance is present. The input clip consists of three sub-sequences, as illustrated in Figure 3(b). Seq 1 and Seq 2 are completely irrelevant, and ahead of them, another sub-sequence called prior context is given as the input, which is exactly the same as Seq 2. Frames marked by black arrows are inputs and those marked by red arrows are expected outputs. There are two training objective: a) to predict 10 future frames of Seq 1; and b) to predict 10 future frames of Seq 2. At the test time, we only evaluate the prediction result of Seq 2. The copy test evaluates the modeling capability of *long-range* video frame relations. A well-designed model should make precise predictions regarding Seq 2, as it has seen all frames of this sequence before. However, this task is difficult for previous LSTM networks. Because Seq 1 is completely irrelevant, the attempt of making predictions of Seq 1 can erase its memory of Seq 2.

The results are presented in the third column (Copy) of Table 1. All baseline models suffer from the influence brought by irrelevant frames in Seq 2 and tend to gradually forget the salient information in the prior context. However, thanks to the eidetic 3D memory, our E3D-LSTM model captures the long-term video frame interactions and performs well in both metrics. A careful inspection of the attention weight shows that the E3D-LSTM model can better attend to useful historical representations across multiple time stamps. The copy test suggests that the E3D-LSTM network is capable of modeling long-range periodical motions effectively.

**Ablation Study.**    We conduct a series of ablation studies and summarize the results in Table 2. First, on the first two rows, we show two alternative 3D-LSTM models with 3D-Convs outside the recurrent unit, including 3D-CNN at Bottom (Figure 1(a)) and 3D-CNN on Top (Figure 1(b)). The performance drop validates the integration of 3D-Convs and RNN units via the eidetic 3D memory. Second, the third baseline method is a special case where all 3D convolutional filters in our model are reduced to 2D. The results demonstrate the effect of capturing local spatiotemporal patterns by the 3D memory within an individual recurrent state. Furthermore, the contribution of the memory attention mechanism can be isolated in the fourth baseline method. Note that all evaluated models are trained with a similar number of parameters for fair comparisons, and the performance gain comes from design options rather than increased model parameters.

## 4.2    FUTURE VIDEO PREDICTION: KTH ACTION

We evaluate the proposed E3D-LSTM model on video prediction of real-world datasets.

**Dataset and Setup.**    The KTH action dataset (Schuldt et al., 2004) contains 25 individuals performing 6 types of actions, including walking, jogging, running, boxing, hand waving and hand clapping. On average, each video clip lasts 4 seconds. We follow the experimental setup in (Villegas et al., 2017) by using person 1-16 for training and 17-25 for testing. Each frame is resized to $128 \times 128$ pixels. We employ the same E3D-LSTM network architecture detailed in Section 4.1. Models are trained to predict next 10 frames from the previous 10 observations. The prediction horizon at the test time is extended to 20 or 40 time stamps.

**Results.**    Table 3 shows quantitative results of the proposed model and state-of-the-art methods. Same as prior work, we use SSIM and PSNR as metrics. Consistent with the observations on the moving MNIST dataset, the E3D-LSTM model performs favorably against the state-of-the-art methods across three settings of predicting future 10 frames, 20 frames, and copy test. These empirical results demonstrate the effectiveness of the E3D-LSTM model for modeling spatiotemporal data.

Figure 4 compares representative generated frames. We select video sequences with relatively complicated spatiotemporal variations (in both moving trajectories and human figure sizes). In the top

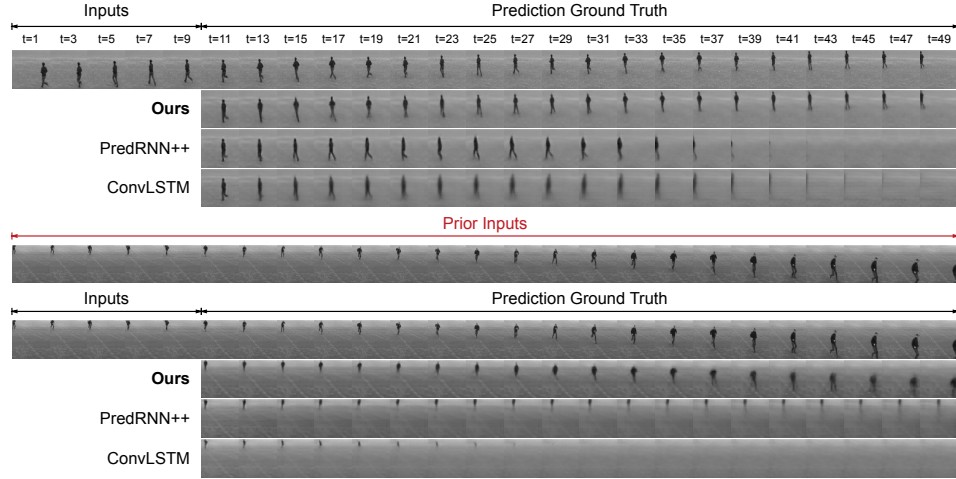

Figure 4: Comparisons of the generated frames on KTH. (Top) predictions of next 40 frames based on 10 previous observations. (Bottom) the copy test that requires to reproducing prior inputs.

Table 3: Quantitative evaluation of different methods on the KTH human action test set. The metrics are averaged over the predicted frames. Higher scores indicate better prediction results.

| MODEL | $10 \rightarrow 20$ | | $10 \rightarrow 40$ | | COPY ($\rightarrow 40$) | |
|---|---|---|---|---|---|---|
| | PSNR | SSIM | PSNR | SSIM | PSNR | SSIM |
| CONVLSTM (SHI ET AL., 2015) | 23.58 | 0.712 | 22.85 | 0.639 | 23.49 | 0.670 |
| DFN (DE BRABANDERE ET AL., 2016) | 27.26 | 0.794 | 23.01 | 0.652 | 23.37 | 0.664 |
| MCNET (VILLEGAS ET AL., 2017) | 25.95 | 0.804 | - | - | - | - |
| FRNN (OLIU ET AL., 2018) | 26.12 | 0.771 | 23.77 | 0.678 | 24.00 | 0.685 |
| PREDRNN (WANG ET AL., 2017) | 27.55 | 0.839 | 24.16 | 0.703 | 24.45 | 0.711 |
| PREDRNN++ (WANG ET AL., 2018B) | 28.47 | 0.865 | 25.21 | 0.741 | 25.90 | 0.759 |
| E3D-LSTM | **29.31** | **0.879** | **27.24** | **0.810** | **30.59** | **0.874** |

half (predicting the next 40 frames based on 10 previous frames), E3D-LSTM predicts more accurate motion trajectories into the future, whereas PredRNN++ and ConvLSTM incorrectly predict the person moving out of the scenes. The lower half shows the *copy test* providing the expected outputs as prior inputs. We directly apply models, which are trained under the first setting, to this test. Without the prior context, it would be difficult to predict human motions for some cases. With prior inputs, E3D-LSTM benefits the most from its memories and responds well to rapid appearance change. In contrast, PredRNN++ and ConvLSTM are not able to capture useful spatiotemporal patterns from distant observations due to the lack of modeling long-term data relations.

### 4.3 A REAL VIDEO PREDICTION APPLICATION: TRAFFIC FLOW PREDICTION

We further evaluate our method on the TaxiBJ dataset, which contains real-world traffic flow data in consecutive heat maps. Predicting urban traffic conditions is a complex setting, as the heat maps are very noisy and we do not have any underlying or additional information that can facilitate this task.

Table 4: Experimental results on the TaxiBJ dataset. We report MSE at every time stamp.

| MODEL | FRAME 1 | FRAME 2 | FRAME 3 | FRAME 4 |
|---|---|---|---|---|
| ST-RESNET (ZHANG ET AL., 2017) | 0.688 | 0.939 | 1.130 | 1.288 |
| VPN (KALCHBRENNER ET AL., 2017) | 0.744 | 1.031 | 1.251 | 1.444 |
| FRNN (OLIU ET AL., 2018) | 0.682 | 0.823 | 0.989 | 1.183 |
| PREDRNN (WANG ET AL., 2017) | 0.634 | 0.934 | 1.047 | 1.263 |
| PREDRNN++ (WANG ET AL., 2018B) | 0.641 | 0.855 | 0.979 | 1.158 |
| E3D-LSTM | **0.620** | **0.773** | **0.888** | **0.984** |

**Dataset and Setup.** The TaxiBJ dataset is collected from the chaotic real-world environment using GPS monitors of taxicabs Beijing. Each frame is a $32 \times 32 \times 2$ heat map. The last dimension denotes

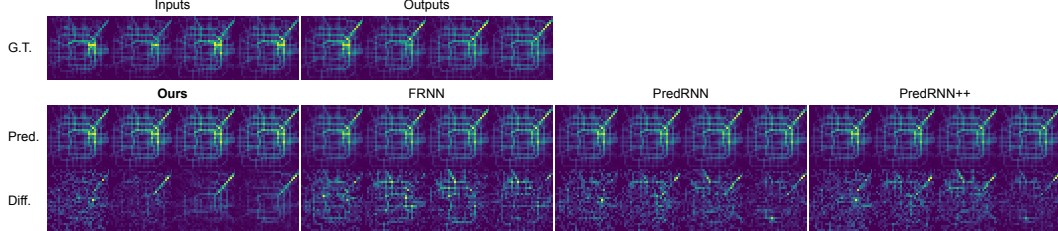

Figure 5: Prediction results on the TaxiBJ traffic flow dataset. For ease of comparison, we visualize the differences between the generated heat maps and their corresponding ground truth heat maps.

the entering and leaving traffic flow intensities at the same area. We split the whole dataset into a training set and a test set as described in (Zhang et al., 2017). We train the networks to predict 4 frames (the next 2 hours) from 4 observations. We use the same network architecture and training setups as the one on Moving MNIST and KTH datasets.

**Results.** We report MSE at every time stamp in Table 4 where lower scores indicate better prediction results. We also show a prediction example in Figure 5. Furthermore, we visualize the differences between the generated heat maps and the ground truth heat maps. Overall, the E3D-LSTM model outperforms the other methods, with the lowest differences intensities in most areas.

## 4.4 EARLY ACTIVITY RECOGNITION: SOMETHING-SOMETHING

To validate that the E3D-LSTM model can learn high-level video representations effectively, we carry out experiments on early activity recognition. The task is to predict an activity category in a video after only observing a fraction of frames. We choose not to evaluate on the full-length video for the activity recognition task, because when a model sees the full-length video, it may make decisions solely based on the scene information, e.g. seeing only the last frame is enough to recognize many actions. As a result, the full-length video task may not align well with our previous video prediction tasks, in which the sequential tendency and causality are important.

**Dataset and Setup.** The something-something dataset (Goyal et al., 2017) is a recent benchmark for activity/action recognition (https://20bn.com/datasets/something-something). We use the standard and official subset which contains $56,769$ short videos for the training set and $7,503$ videos for the validation set on $41$ action categories. The video length ranges between 2 and 6 seconds with 24 fps. We adopt the early activity recognition setting (Ma et al., 2016; Zeng et al., 2017; Zhou et al., 2018), where a model predicts an action type after observing the first 25% or 50% frames of each video. As these actions appear in diverse scenes and involve interaction with different objects, it is challenging to predict actions even for humans (See Figure 6). There are only subtle differences between some actions in this dataset, such as "Poking a stack of [Something] so that the stack collapses" versus "Poking a stack of [Something] without the stack collapsing", or "Pouring [Something] into [Something]" versus "Trying to pour [Something] into [Something], but missing so it spills next to it". To make a correct prediction, a model needs to exploit spatiotemporal cues to understand the subtle differences between actions. Namely, one can evaluate the model effectiveness for high-level video tasks. Recognizing early action accurately requires predictions into future frames, which can only be achieved using an effective model based on historical observations.

**Hyper-parameters and Baselines.** We use the architecture illustrated in Figure 1(c) as our model, which consists of 2 layers of 3D-CNN encoders, 4 layers of E3D-LSTMs, and 2 layers of 3D-CNN decoders. The 3D-CNN encoders take 4 consecutive $224 \times 224$ raw frames, encode them into $2 \times 56 \times 56 \times 64$ feature maps at each time stamp, and then feed them into E3D-LSTM. Each encoder layer has $64$ filters (the filter dimensions are $2 \times 5 \times 5$). We use the same hyper-parameters for E3D-LSTMs as for video prediction. The decoder layers map the output of E3D-LSTMs back to RGB space, which is an $1 \times 3$ matrix, predicting the next frame following the inputs. We train the network to predict the next 10 frames using the front 25% or 50% frames of the video. Note that we do not extend any predictive states into the future at the test time. For both training and testing, we concatenate hidden representations of the top recurrent units with respect to the last 16 input time stamps (considering the first 25% video snippets usually have about 20 to 30 frames), and feed them into the classifier for activity recognition. The classifier contains 2 layers of 3D-Convs with $128$ filters (filter dimensions: $2 \times 3 \times 3$, filter strides: $2 \times 2 \times 2$) followed by a $2 \times 2 \times 2$ pooling layer. They transform the concatenated recurrent features from $16 \times 56 \times 56 \times 64$ to $1 \times 7 \times 7 \times 128$,

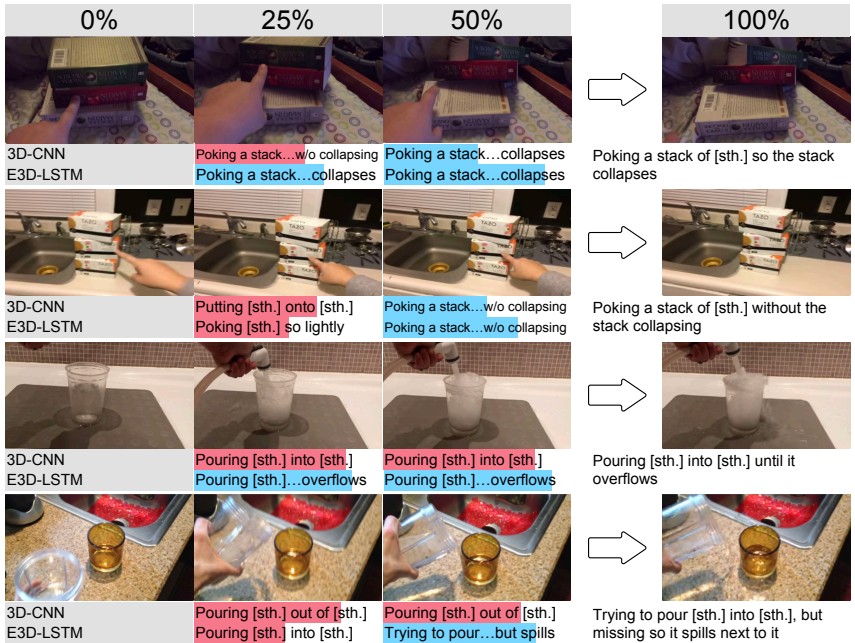

Figure 6: Early activity recognition results given the first 25% and 50% frames of videos on the Something-Something validation set. The blue bars indicate making correct classifications and the red bars are incorrect results. The length of the bar denotes the confidence of the result.

Table 5: Early activity recognition accuracy on the 41-category subset of Something-Something.

| MODEL | FRONT 25% | FRONT 50% |
|---|---|---|
| 3D-CNN | 9.11 | 10.30 |
| SEPARABLE-CNN: SEPARABLE-CONV AT BOTTOM | 8.94 | 9.62 |
| (2+1)D-CNN: SEPARABLE-CONV ON TOP | 9.08 | 10.17 |
| E(2+1)D-LSTM: SEPARABLE INSIDE UNITS | 12.45 | 19.86 |
| **E3D-LSTM** | **14.59** | **22.73** |

then pass them to a 512-channel fully-connected layer followed by a 41-way classification. We also exploit the self-supervised auxiliary learning approach and train the model with an objective function in Equation 4. We set $\lambda(i)$ in Equation 5 to 10 in the beginning ($i = 0$), and decrease it with a speed of $2 \times 10^{-5}$ per iteration, lower bounded by $\eta = 0.1$.

We evaluate the E3D-LSTM model against the state-of-the-art feed-forward 3D-Conv architectures including C3D/I3D (Diba et al., 2016; Carreira & Zisserman, 2017), Separable 3D-CNN (Xie et al., 2018; Qiu et al., 2017) and (2+1)D-CNN (Tran et al., 2018). These networks achieve the state-of-the-art results on the UCF-101 and Kinetics benchmark datasets for action recognition. For fair comparisons, we train these baseline models using similar backbones to the E3D-LSTM network.

**Results.** Table 5 shows the classification accuracy of the E3D-LSTM network against the state-of-the-art feed-forward 3D-CNNs. The E3D-LSTM model performs favorably against the other methods in two settings of using the first 25% and 50% frames, showing its effectiveness in learning high-level spatiotemporal representations. Figure 6 shows two pairs of video activities that are easy to confuse, especially with such limited observations. For instance, our model correctly forecasts the collapse of books, while only a tendency of it has been shown explicitly within the first 25% frames. This reasoning ability comes from the integrated design of our model to capture both short-term motions and long-term dependencies. On the other hand, as the feed-forward 3D-CNN models long-term relations by sampling and assembling, it does not perform well in finding the temporal dependencies between cause and effect. We note that Zhou et al. (2018) introduced a feed-forward CNN model and also reported early recognition results on the same dataset. It is not meaningful to compare these two methods in terms of accuracy as our model is trained only using 25%-50% frames of a video instead of the entire video in (Zhou et al., 2018). Moreover, the two methods are trained using different backbone networks and different splits of datasets.

Table 6: Ablation study of early activity recognition on the Something-Something dataset.

| MODEL | FRONT 25% | FRONT 50% |
|---|---|---|
| BASELINE 1: 3D-CNN AT BOTTOM (FIGURE 1(A)) | 10.28 | 16.05 |
| BASELINE 2: 3D-CNN ON TOP (FIGURE 1(B)) | 9.63 | 14.82 |
| BASELINE 3: OURS W/O 3D CONVOLUTIONS | 9.58 | 13.92 |
| BASELINE 4: OURS W/O MEMORY ATTENTION | 11.39 | 18.84 |
| E3D-LSTM | **14.59** | **22.73** |

Table 7: Accuracy comparisons of different training strategies on the Something-Something dataset.

| MODEL | FRONT 25% | FRONT 50% |
|---|---|---|
| TRAINED ONLY ON THE PRIMARY CLASSIFICATION TASK | 13.78 | 20.91 |
| PRE-TRAINED ON THE AUXILIARY TASK | 14.00 | 22.15 |
| TRAINED ON BOTH TASKS WITH A FIXED LOSS RATIO | 13.57 | 20.46 |
| E3D-LSTM (WITH SELF-SUPERVISED AUXILIARY LEARNING) | **14.59** | **22.73** |

A number of recent studies show that separating temporal and spatial convolution operations in a 3D-CNN model leads to better results (Xie et al., 2018; Qiu et al., 2017; Tran et al., 2018). This observation is validated by our results shown in Table 5. However, it seems counter-intuitive since such separation leads to a pseudo-3D convolution, in which spatial and temporal filters are independent. Interestingly, such separation in our model leads to performance loss, suggesting the 3D convolution in the E3D-LSTM jointly captures the temporal and spatial information.

**Ablation Study.** We conduct similar ablation studies as in Section 4.1 and summarize the results in Table 6. The results from the first two rows show that our deeper integration of 3D-Convs inside RNNs is helpful not only for pixel-level video prediction, but also for high-level activity recognition. The results on rows 3 and 4 show the contribution of the two important components in the proposed Eidetic 3D LSTM: a) 3D convolution features, and b) memory attention mechanism. Both components are useful and important for modeling spatiotemporal data effectively. Table 7 shows applying self-supervised training in different settings. The proposed self-supervised auxiliary learning approach performs better than other alternatives, including using video prediction models as network initialization, or training the model under these two tasks with a fixed objective function ratio.

We enable online early activity recognition by making the classifier only depend on a concatenation of the last 5 recurrent output states. Using Equation 1, we fix the length of the attended memory states by setting $\tau$ to 5. Such settings are applied to both training and testing. Table 8 shows the experimental results. Despite the slight decrease of accuracy, it enables an online prediction.

Table 8: Online early recognition accuracy: the classifier is built on the last 5 recurrent output states.

| MODEL | FRONT 25% | FRONT 50% |
|---|---|---|
| TRAINED ONLY ON THE PRIMARY CLASSIFICATION TASK | 13.49 | 18.94 |
| E3D-LSTM (WITH SELF-SUPERVISED AUXILIARY LEARNING) | **14.30** | **20.85** |

## 5 CONCLUSION

Spatiotemporal predictive learning has shown significant improvements in a variety of applications, such as weather forecasting, traffic flow prediction, and physical interaction simulation. Although considered to be a promising self-supervised feature learning paradigm, it seldom shows its effectiveness beyond video prediction. In this paper, we presented the E3D-LSTM model based on 3D convolutional recurrent units for this task. In this model, we integrated 3D-Convs into state transitions to perceive short-term motions and designed a memory attentive module controlled by recurrent gates to capture the long-term video frame interaction. Experimental results demonstrate that the E3D-LSTM model performs favorably against the state-of-the-art methods on video prediction and early activity recognition tasks.

ACKNOWLEDGMENTS

We would like to thank anonymous reviewers for useful comments. Mingsheng Long was supported by National Natural Science Foundation of China (61772299, 71690231).

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

## A  KEY EQUATIONS OF SPATIOTEMPORAL LSTM

Operations inside a Spatiotemporal LSTM unit at time stamp $t$ and layer $k$ are shown as follows:

$$i_t = \sigma(W_{xi} * \mathcal{X}_t + W_{hi} * \mathcal{H}_{t-1}^k + b_i)$$
$$g_t = \tanh(W_{xg} * \mathcal{X}_t + W_{hg} * \mathcal{H}_{t-1}^k + b_g)$$
$$f_t = \sigma(W_{xf} * \mathcal{X}_t + W_{hf} * \mathcal{H}_{t-1}^k + b_f)$$
$$i'_t = \sigma(W'_{xi} * \mathcal{X}_t + W_{mi} * \mathcal{M}_t^{k-1} + b'_i)$$
$$g'_t = \tanh(W'_{xg} * \mathcal{X}_t + W_{mg} * \mathcal{M}_t^{k-1} + b'_g)$$
$$f'_t = \sigma(W'_{xf} * \mathcal{X}_t + W_{mf} * \mathcal{M}_t^{k-1} + b'_f) \tag{6}$$
$$\mathcal{C}_t^k = i_t \odot g_t + f_t \odot \mathcal{C}_{t-1}^k$$
$$\mathcal{M}_t^k = i'_t \odot g'_t + f'_t \odot \mathcal{M}_t^{k-1}$$
$$o_t = \sigma(W_{xo} * \mathcal{X}_t + W_{ho} * \mathcal{H}_{t-1}^k + W_{co} * \mathcal{C}_t^k + W_{mo} * \mathcal{M}_t^k + b_o)$$
$$\mathcal{H}_t^k = o_t \odot \tanh(W_{1\times 1} * [\mathcal{C}_t^k, \mathcal{M}_t^k]),$$

where $\sigma$ is the sigmoid function, $*$ is the convolution operator, and $\odot$ denotes the Hadamard product. There are four inputs: $\mathcal{X}_t$, the raw frame or hidden states from the previous layer; $\mathcal{M}_t^{k-1}$, the previous spatiotemporal memory; $\mathcal{H}_{t-1}^k$ and $\mathcal{C}_{t-1}^k$, the previous hidden states and memory states. Two sets of gate structures, including input gate $i_t$ and $i'_t$, forget gate $f_t$ and $f'_t$, as well as the output gate $o_t$, control the information flow in space-time domain. All of them can be presented by $\mathbb{R}^{H \times W \times C}$ dimensional tensors, where the first two dimensions are the width and height of feature maps, and the last one is the number of feature map channels.

