# OpenReview forum: "Eidetic 3D LSTM: A Model for Video Prediction and Beyond"
_ICLR.cc/2019/Conference_

### Official Review · AnonReviewer2 · 2018-10-29
**Nice experiments although it lacks a bit of novelty**

**Rating:** 7
**Confidence:** 4

**Review:**

# 1. Summary
This paper presents a model for future video prediction, which integrates 3D convolutions into RNNs. The internal operations of the RNN are modified by adding historical records controlled via a gate-controlled self-attention module. The authors show that the model is effective also for other tasks such as early activity recognition.

Strengths:
* Nice extensive experimentation on video prediction and early activity recognition tasks and comparison with recent papers
* Each choice in the model definition are motivated, although some clarity is still missing (see below)

Weaknesses:
* Novelty: the proposed model is a small extension of a previous work (Wang et al., 2017)


# 2. Clarity and Motivation
In general, the paper is clear and general motivation makes sense, however some points need to be improved with further discussion and motivation:

A) Page 2 “Unlike the conventional memory transition function, it learns the size of temporal interactions. For longer sequences, this allows attending to distant states containing salient information”: This is not obvious. Can the authors add more details and motivate these two sentences? How is long-term relations are learned given Eq. 1?
B) Page 5 “These two terms are respectively designed for short-term and long-term video modeling”: How do you make sure that Recall(.) does not focus on the short-term modeling instead? Not clear why this should model long-term relations.
C) Page 5 and Eq 1: motivation why layer norm is required when defining C_t^k is not clear
D) What if the Recall is instead modeled as attention? The idea is to consider only C_{1:t-1}^k (not consider R_t) and have an attentional model that learn what to recall based only on C. Also, why does Recall need to depend on R_t?
E) Page 5 “to minimize the l1 + l2 loss over every pixel in the frame”: this sentence is not clear. How does it relate to Eq. 2?


# 3. Novelty
Novelty is the major concern of this paper. Although the introduced new concepts and ideas are interesting, the work seems to be an extension of ST-LSTM and PredRNN where Eq 1 is slightly modified by introducing Recall.
In addition the existing relation between the proposed model and ST-LSTM is not clearly state. Page 2, first paragraph: here the authors should state that model is and extension of ST-LSTM and highlight what are the difference and advantage of the new model.


# 4. Significance of the work
This paper deals with an interesting and challenging topic (video prediction) as well as it shows some results on the early activity recognition task. These are definitively nice problem which are far to be solved. From the application perspective this work is significant, however from the methodological perspective it lacks a bit of significance because of the novelty issues highlighted above.


# 5. Experimentation
The experiments are robust with nice comparisons with recent methods and ablation study motivating the different components of the model (Table 1 and 2). Some suggested improvements:

A) Page 7 “Seq 1 and Seq 2 are completely irrelevant, and ahead of them, another sub-sequence called prior context is given as the input, which is exactly the same as Seq 2”: The COPY task is a bit unclear and need to be better explained. Why are Seq. 1 and 2 irrelevant? I would suggest to rephrase this part.
B) Sec. 4.2, “Dataset and setup”: which architecture has been used here?
C) Sec. 4.3, “Hyper-parameters and Baselines“: the something-something dataset is more realising that the other two “toy” dataset. Why did the authors choose to train a 2 layers 3D-CNN encoders, instead of using existing pretrained 3D CNNs? I would suspect that the results can improve quite a bit.


# 6. Others
* The term “self-supervised auxiliary learning” is introduced in the abstract, but at this point it’s meaning is not clear. I’d suggest to either remove it or explain its meaning.
* Figure 1(a): inconsistent notation with 2b. Also add citation (Wang et al., 2017) since it ie the same model of that paper

-------
# Post-discussion
I increased my rating: even if novelty is not high, the results support the incremental ideas proposed by the authors.

---

> ### Author Response · Authors · 2018-11-26
> **Responses to AnonReviewer2**
>
>
> Q7 Novelty: the proposed model is a small extension of a previous work [Wang et al., 2017]
> >> Please see the answer to Q3 above (the novelty of the paper)
>
> Q8 Clarity and Motivation:
>
> A) Page 2...How long-term relations are learned given Eq. 1?
> >> Different from standard LSTMs, we are motivated by modeling the long-term relations across frames.  The long-term relations are learned by the RECALL function in Eq. 1, whose inputs are the historical memory states C_{t-\tau:t-1}^k (in particular, we use C_{1:t-1}^k for most experiments in this work). The RECALL function queries useful information from C_{t-\tau:t-1}^k using R_t. We have clarified this point in the paper (Page 4).
>
> B) Page 5...Not clear why Recall(.) should model long-term relations.
> >> The RECALL function enables an adaptive learning of short-term and long-term modeling. More specifically, in Eq. 1, C_{t-1}^k is added to C_t^k via a short-cut connection controlled by the forget gate. Intuitively, it conveys short-term information, thus allowing the RECALL function to focus on long-term relations. Empirically, the COPY task verifies that our model could make use of information from the distant memory states when future predictions are severely dependent on the distant past.
>
> C) Eq 1: why layer norm is required when defining C_t^k is not clear.
> >> We use the layer normalization technique to mitigate the covariant shift and stabilize the training process, as it has been commonly used in RNNs. We have made it clear in the paper.
>
> D) What if the Recall is instead modeled as attention?
> >> Making the RECALL function solely based on memory states C will make the relations between C_{t-1}^k and itself (or the relations between very short-term memory states) dominate the result of RECALL(.). Thus, we encode X_t and H_{t-1}^k into R_t, and use it as the query of the attentive RECALL function.
>
> E) Page 5 “to minimize the l1 + l2 loss over every pixel in the frame” is not clear.
> >> We use different objective functions for different tasks:
> 1. Video prediction: L1 + L2 loss.
> 2. Early activity recognition: Eq. 3 in the revised paper.
>
> Q9 Experiments:
>
> A) Page 7...Why are Seq. 1 and Seq. 2 irrelevant?
> >> We have rephrased this part in Page 6. Basically, the COPY task is to evaluate whether our model could recall useful information from the distant memory states. A well-performed predictive model should make precise predictions regarding Seq 2, as it has seen all frames of this sequence before. But this task is difficult for previous LSTM networks. Because the Seq 1 is totally irrelevant, making predictions of Seq 1 will erase its memory of Seq 2.
>
> B) Sec. 4.2, “Dataset and setup”: which architecture has been used here?
> >> We have made it clear that the architecture for KTH is exactly the same as that for the Moving MNIST.
>
> C) Sec. 4.3...the something-something dataset is more realising than the other two “toy” dataset. Why did the authors choose to train a 2 layers 3D-CNN encoders, instead of using existing pretrained 3D CNNs?
> >> In this paper, our goal is to explore a generic method that can infer the action tendency and intentions from sequential video frames. We show that in a fair setting (the same training set and similar #learnable parameters), the improvements of our work come from a better model to capture and predict low-level video data trends, along with a better understanding of high-level actions.
> Although using the 3D-CNN model pre-trained on video datasets may improve the results, it also makes fair comparisons among all methods very tricky. First, suppose a model improves the results; it is less clear whether it is because the model learns a better representation on the pretrained data, or it is actually better in modeling the target dataset. Second, due to the domain difference, it is hard to select which pretrained models (e.g. Sports1M or Kinetics) to use on which dataset, and the pretrained model works on one dataset (e.g. something-something) may not work well on another dataset(e.g. KTH). These issues can result in a lengthy and unclear experimental section.

---

> > ### Comment · AnonReviewer2 · 2018-11-28
> > **Comment to authors**
> >
> > Q7 (novelty)
> > 1) It may be the first work using 3d convolutions in RNNs, however there is already a previous work using 2d convolution in RNNs: "Convolutional LSTM Network: A Machine Learning Approach for Precipitation Nowcasting, NIPS 2015.".
> >
> > Q8 E) Please add this info to the paper.
> >
> > Q9 C) It would have been interesting to see an experiment with one of these pre-trained models, because the used 2-layer network might be not be able to learn good features for the task.
> >
> > Overall novelty is still not fully convincing, however the results support the incremental ideas proposed by the authors.

---

### Official Review · AnonReviewer1 · 2018-11-01
**well-written, well-experimented paper with limited novelty**

**Rating:** 7
**Confidence:** 4

**Review:**

The paper proposes a spatiotemporal modeling of videos based on two currently available spatiotemporal modeling paradigms: RNNs and 3D convolutions. The main idea of this paper is to get the best world of both in a unified way. The method first encodes a sequence of frames using 3D-conv to capture short-term motion patterns, passes it to a specific type of LSTM (E3D-LSTM) which accepts spatiotemporal feature maps as input. E3D-LSTM captures long-term dependencies using an attention mechanism. Finally, there are 3D-conv based decoders which receive the output of E3D-LSTM and generate future frames. The message of the paper, I believe, is that 3D-conv and RNNs can be integrated to perform short and long predictions. They show in the experiments how the model can remember far past for reasoning and prediction.
The nice point of the method is that it is heavily investigated through experiments. It's evaluated on two datasets, with ablation studies on both. Moreover, the paper is well-written and clear. technically, the paper seems correct.
However, my only big concern is about the limited novelty of the method. E3D-LSTM is the core of the novelty, which is basically an LSTM with extra gate, and attention mechanism.

other comments:
- As the method by essence is a spatiotemporal learning model, why the method is not evaluated on full-length videos of the something-something dataset for classical action classification task, in order to compare it with the full architecture of I3D, or S3D?

- While the paper discusses self-supervised learning, I would suggest showing its benefit on online action recognition task. One without frame-prediction loss and one with.

- the something-something dataset has 174 classes, how was the process of selecting 41 classes out of it?

---

> ### Author Response · Authors · 2018-11-26
> **Responses to AnonReviewer1**
>
>
> Q3: Concern about the novelty: my only big concern is about the limited novelty of the method. E3D-LSTM is the core of the novelty, which is basically an LSTM with extra gate, and attention mechanism.
>
> The concern on limited novelty is mainly due to the seeming similarity to the prior work [Wang et al., 2017]. Below we clarify the differences to the prior work:
>
> 1. Our paper is one of, if not the first, work to systematically explore 3D convolutions **inside** the RNN. More importantly, it is the first to show a carefully designed method achieves the state-of-the-art results on several public benchmarks. The improvements are otherwise not shown for any known combination of 3D convolutions and RNN.
>
> 2. Our technical difference to the existing work includes:
> a. we study where to apply the 3D info. For example, combine 2D or 3D inputs (see Figure 1), inflate the LSTM cell to 3D (see Figure 2b), or separate the 3D convolutions in the input and LSTM cell (see Table 4).
> b. we propose how to effectively embed the 3D convolution inside the LSTM (i.e. we introduce a new recall gate in Equation 1 for the 3D-memory transition inside the LSTM).
>
> Among the recent advances in deep learning, many great models appear to be similar to prior work (e.g. ResNet and Highway Network, ConvLSTM and LSTM, C3D/I3D CNN and 2D CNN). However, it is not true as the devil is in the important details. Similarly, we build upon prior work, make only necessary, yet important, model designs, and validate the necessities with ablation studies to demonstrate their merits.  Our designs are driven by a clear motivation, innovative thinkings, and validated by extensive experiments (as agreed by all reviewers).
> We hope this can resolve the concern on novelty.
>
> Q4: As the method by essence is a spatiotemporal learning model, why the method is not evaluated on full-length videos of the something-something dataset?
>
> The main reason is that predicting on the full-length video may not align well with our topic. A typical video classification model which can see full-length videos may make decisions mainly depending on the scene information. As shown in Fig. 5, suppose the tasks is to predict a category “Poking a stack of [Something], so the stack collapses”. The problem would be very simple as long as the model sees the last frame which shows the outcome of the action.
>
> In contrast, the early activity recognition task makes the model have no other choices but to depend on an inference of the action intentions when making decisions. It aligns well with the video prediction task, in which the sequential tendency and causality are important.
>
> We notice that it would be more accurate to claim our model as a spatiotemporal predictive learning model, rather than a broad “spatiotemporal learning model”. We have revised that in the paper.
>
> Q5: Show the benefit on online action recognition task.
>
> As suggested, we have added online early activity recognition by making the classifier only depend on a concatenation of the recurrent outputs regarding the last 5 timestamps. As such, the historical recurrent states are only kept for 5 timestamps and then discarded. In particular, we apply a sliding window of limited length to the inputs of the Recall gate, using $C_{t-5:t-1}$ instead of $C_{1:t-1}$ in Eq. 1. Experimental results are shown in Table 7. Despite the slightly decreased accuracy, applying the sliding window on the Recall gate improves the scalability of E3D-LSTM.
>
> Q6: How was the process of selecting 41 classes out of the something-something dataset?
>
> In the original paper [Goyal et al. 2017] of the Something-Something dataset, the  41 classes (in Table 7) are listed as a standard and official dataset setting. This split contains 56k video clips for training and 7.5k for validation and is large enough and meanwhile computational convenient to compare a variety of baseline methods. We have clarified this point in the paper (Page 9).

---

> > ### Comment · AnonReviewer1 · 2018-11-30
> > **comment to authors**
> >
> > Q3:
> > As R1 said, there are works integrating 2d convolution and RNNs, like "VideoLSTM convolves, attends and flows for action recognition". still, novelty is not convincing.
> >
> > Q4: A typical video classification model which can see full-length videos may make decisions mainly depending on the scene information.
> > I understand this paper aims to predict the future. however, "Zhou et. al, Temporal Relational Reasoning in Videos" show that for recognizing actions in something-something dataset, scene clues are not enough and modeling temporal dependencies are important. so a classical classification problem on this dataset makes sense.
> >
> > Though novelty is still not fully convincing, the paper can shed insights into the topic.

---

### Official Review · AnonReviewer3 · 2018-11-02
**The authors propose a futurue video prediction model based on recurrent 3D-CNNs and propose a novel memory mechanism (Eidetic Memory) to capture long term relationships inside the recurrent layer itself. They obtain surpass the state of the art on two commonly used, (relatively) simple benchmark video prediction datasets. They further apply their model to early action recognition, performing an ablation study to evaluate the strengths of each model building block.**

**Rating:** 7
**Confidence:** 5

**Review:**

AFTER REBUTTAL:

This is an overall good work, and I do think proves its point. The results on the TaxiBJ dataset (not TatxtBJ, please correct the name in the paper) are compelling, and the concerns regarding some of the text explainations have been corrected.

-----

The proposed model uses a 3D-CNN with a new kind of 3D-conv. recurrent layer named E3D-LSTM, an extension of 3D-RCNN layers where the recall mechanism is extended by using an attentional mechanism, allowing it to update the recurrent state not only based on the previous state, but on a mixture of previous states from all previous time steps.

Pros:
The new approach displays outstanding results for future video prediction. Firstly, it obtains better results in short term predictions thanks to the 3D-Convolutional topology. Secondly, the recall mechanism is shown to be more stable over time: The prediction accuracy is sustained over longer preiods of time (longer prediction sequences) with a much smaller degradation. Regarding early action recognition, the use of future video prediction as a jointly learned auxiliary task is shown to significantly increase the prediction accuracy. The ablation study is compelling.

Cons:
The model does not compare against other methods regarding early action recognition. Since this is a novel field of study in computer vision, and not too much work exists on the subject, it is understandable. Also, it is not the main focus of the work.

In the introduction, the authors state that they account for uncertainty by better modelling the temporal sequence. Please, remove or rephrase this part. Uncertainty in video prediction is not due to the lack of modelling ability, but due to the inherent uncertainty of the task. In real world scenarios (eg. the KTH dataset used here) there is a continuous space of possible futures. In the case of variational models, this is captured as a distribution from which to sample. Adversarial models collapse this space into a single future in order to create more realistic-looking predictions. I don't believe your approach should necessarily model that space (after all, the novelty is on better modelling the sequence itself, not the possible futures, and the model can be easily extended to do so, either through GANs or VAEs), but it is important to not mislead the reader.

It would have been interesting to analyse the work on more complex settings, such as UCF101. While KTH is already a real-world dataset, its variability is very limited: A small set of backgrounds and actions, performed by a small group of individuals.

---

> ### Author Response · Authors · 2018-11-26
> **Responses to AnonReviewer3.**
>
>
> Q1: In the introduction, the authors state that they account for uncertainty by better modeling the temporal sequence...
>
> We have rephrased this expression for clarity in the revised paper (Page 2). Below is a copy: Future prediction errors of an imperfect model can be categorized by two factors: a) the “systematic errors” caused by a lack of modeling ability to the deterministic variations; b) the stochastic, inherent uncertainty of the future. We aim to minimize the first factor in this work.
>
> Q2: Analyze the work in more complex settings.
>
> We have experimented with the Something-Something dataset for video prediction, but the generated frames are not satisfying even when integrated with adversarial training and variational methods. The results are not surprising as the number of training samples is too limited to capture the diverse scenes of real-world videos (due to the illumination, occlusion, camera motion, to name a few). This makes future prediction considerably difficult for all existing methods, including ours. Exploring very complex datasets will be an interesting future research direction for this task.
>
> However, as R3 suggested, we further evaluate our method on a real-world dataset for traffic flow prediction, i.e., TaxtBJ. In this dataset, traffic flows (in consecutive heat maps) are collected from the chaotic real-world environment. Predicting urban traffic conditions is a complex setting, as the heat maps are very noisy and we do not have any corresponding, underlying, additional information. Implementation details and empirical results can be found in Appendix B. We train the networks to predict 4 frames (the next 2 hours) from 4 observations and report MSE at every timestamp. As shown, our method achieves the state-of-the-art result in Table 8 and generates the most accurate predictions in Fig. 6.

---

### Author Response · Authors · 2018-11-26
**We thank reviewers for the valuable comments.**

We thank reviewers for the valuable comments. Based on the reviews, we make the following changes (we mark these changes in blue in the revised paper):

1. As suggested by R1, we enable our method to perform the online recognition tasks and compare our online model with and without the frame-prediction loss in Table 7.

2. As suggested by R3, we add an additional real-world dataset on traffic flow prediction and evaluate our method under this complex setting. The results are presented in Appendix B.

3. We rephrase/clarify all of the points raised by the reviewers.

We will address all questions in the individual replies.

---

### Meta-Review · Area_Chair1 · 2018-12-13
**Well executed exploration of a 3D CNN LSTM method**

**Confidence:** 4
**Recommendation:** Accept (Poster)

**Metareview:**

Strengths: Strong results on future frame video prediction using a 3D convolutional network. Use of future video prediction to jointly learn auxiliary tasks shown to to increase performance. Good ablation study.

Weaknesses: Comparisons with older action recognition methods. Some concerns about novelty, the main contribution is the E3D-LSTM architecture, which R1 characterized as an LSTM with an extra gate and attention mechanism.

Contention: Authors point to novelty in 3D convolutions inside the RNN.

Consensus: All reviewers give a final score of 7- well done experiments helped address concerns around novelty. Easy to recommend acceptance given the agreement.